

# Relationship between the average slope in the active commuting to and from school and fitness in adolescents: the mediator role of fatness

Pedro Antonio Sánchez Miguel[1], David Sánchez Oliva[2], Mikel Vaquero Solís[1], J. J. Pulido[2] and Miguel Angel Tapia Serrano[1]

[1] Department of Didactics of Musical, Plastic and Body Expression. Faculty of Teaching Training, University of Extremadura, Cáceres, Spain
[2] Department of Didactics of Musical, Plastic and Body Expression. Faculty of Sports Sciences, University of Extremadura, Cáceres, Spain

Corresponding authors
Pedro Antonio Sánchez Miguel,
pesanchezm@unex.es
Miguel Angel Tapia Serrano,
matapiase@unex.es

## ABSTRACT

Active commuting to and from school (ACS) has been recognized as a potential tool to improve physical fitness. Thus, this study aims to test the relationships between the average slope in the ACS and physical fitness, as well as to verify the mediator role of fatness in the relationship between average slope and physical fitness. A total of 257 participants, 137 boys and 120 girls, from 22 schools belonged to first and second High School grades participated in this study. Based on self-reported measure and Google Earth, participants were grouped into the active commuter (number of trips was ≥5, and the time of the trip was ≥15 min), mixed commuter (number of weekly trips was <5, and the time spent on the trip was <15 min) and passive commuter groups (those who reported traveling regularly by car, motorcycle, or bus). Specifically, in the active commuter group, a positive association between the average slope in the ACS with fatness was found, which in turn was positively related to strength lower limbs and cardiorrespiratory fitness. The average slope was not significantly associated with physical fitness indicators. Furtheremore, fatness did not mediate the relatihsionship between average slope and physical fitness. This research concluded positive associations between average slope and the body fat in the ACS. The tendency of findings signal that the average slope should be taken into account along with the distance, time and frequency of the active commuting.

## INTRODUCTION

Active commuting to and from school (ACS) can be defined as the mode of commuting by which children or adolescents cover the distance between home and school, using a way that does not involve motorized vehicles, such as walking or cycling (*Chillón et al., 2011*; *Larouche et al., 2014*). In contrast, passive commuting refers to the use of motorized vehicles as a way of transport, such as car, bus, subway, train, motorcycle, or others (*Villa-González et al., 2016a*). ACS has been recognized as a potential tool to increase daily

physical activity or improve physical fitness among adolescents (*Larouche et al., 2014*; *Martin et al., 2016*; *Muntaner-Mas et al., 2018*; *Slingerland, Borghouts & Hesselink, 2012*).

*Andersen et al. (2011)* tested the benefits of this increase in adolescents, reporting lower levels of body fat and a lower probability of heart disease. In this regard, *Ramírez-Vélez et al. (2017)* showed that ACS was associated with greater physical fitness. Despite the clear benefits of physical activity and active school travel, recent evidence across 38 countries from six continents showed that only 20–39% of children and youth were adequately physically active for health (*Tremblay et al., 2016*). Recent researches have indicated a decrease of ACS in children and adolescence in worldwide (*Chillón et al., 2013*; *Ministry of Transport, 2018*; *Yang et al., 2016*). Due to this decrease, interest in the potential correlates of ACS has increased in the last years (*Sallis et al., 2016*; *Rodríguez-Rodríguez et al., 2019*). Trying to find possible related factors, the systematic review conducted by *Ikeda et al. (2018)* found positive associations between ACS and perceptions of safety, walkability and neighborhood social interaction, but any relationship in the average slope in the ACS at school was found. In this regard, it has been shown that sections with an average slope are characterized by producing internal mechanical work, involving greater muscular activity and increasing the linear energy demand (*Ehrström et al., 2017*; *Vernillo et al., 2016*).

In accordance to this issue, previous studies (*Chillón et al., 2011*; *Muntaner-Mas et al., 2018*) has highlighted the need to objectively quantify the environmental characteristics of the routes because of the great variety of factors that influence active commuting (for example, environmental characteristics, economic level of families, use of information and communication technologies, family aspects, cultural patterns…). However, few studies have focused on objectively quantifying physical environmental attributes as the average slope during ACS at school (*Batista, Cooper & Audrey, 2018*). The Geographic Information Systems can be as a solution for this problem because it has been showed that are valid and reliable instruments to assess the built environment (*Veitch et al., 2017*) or connectivity between streets and pedestrian lanes (*McCormack & Shiell, 2011*; *Muntaner-Mas et al., 2018*).

In this line, as was previously indicated, it has been demonstrated that ACS can be a key form of habitual physical activity (*Larouche et al., 2014*; *Sallis et al., 2016*). Furthermore, a positive association with cardiorespiratory fitness in adolescents was shown in different researches (*Lubans et al., 2011*; *Muntaner-Mas et al., 2018*). Nevertheless, there was a negative relationship between ACS and body fat (*Carson et al., 2016*; *Mytton, Panter & Ogilvie, 2016*). Therefore, an indirect effect of fatness between relation positive slope of ACS and physical fitness could be expected. However, up to our knowledge, there are none study that evaluated if adiposity could affect in the associations between ACS and fitness components. Hence, the present investigation proposed as aims: (1) to determine the relationship between the percentage of the average slope in the ACS with physical fitness and (2) to verify if the fatness is a mechanism to explain the association between average slope and physical fitness. In accordance with the aims indicated, several hypotheses are suggested. The first hypothesis suggested that the active commuters would

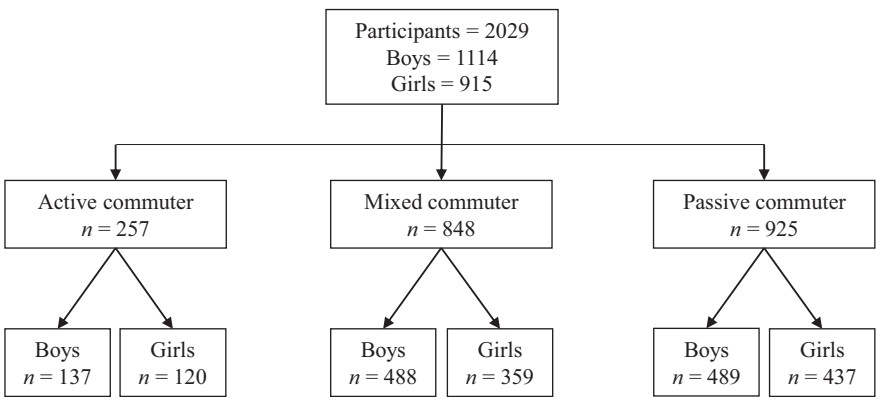

**Figure 1 Flow chart of participants according to the mode of travel.**

have a better physical fitness. The second hypothesis proposed that the active commuters would have a lesser fatness and as a consequence would have a better physical fitness.

## MATERIALS AND METHODS

### Design and participants

The present cross-sectional study was developed in the framework of the promote physical activity and healthy habits in adolescent's project developing in the extracurricular time (ClinicalTrials: NCT03974607, registered 19 August 2019). Sample selection was purposeful. The baseline data were collected from March 2018 to June 2019 in the Extremadura (Spain). For the present study, a total of 22 schools were assessed, with a final sample of 257 adolescents aged 13–16 years (13.20 ± 0.05 years) whom 137 were boys (13.25 ± 1.03 years) and 120 girls (13.15 ± 0.81 years) were included with complete baseline data for anthropometric measures, physical fitness and ACS variables (mode, number of trips, distance, time and average school). In Fig. 1, the sample selection diagram can be proved.

Parents and school supervisors were informed by letter about the nature and purpose of the study, and written informed consent was provided. The study was conducted in accordance with the Declaration of Helsinki, and the protocol was approved by the Ethics Committee of the University of Extremadura (892016).

### Instruments

#### Anthropometrical data

Adiposity parameters were evaluated by anthropometric techniques following standardized procedures. All adiposity measurements were performed twice and the average was recorded. Triceps and subscapular skin-fold thickness were evaluated according to standard procedures (Harpenden; range, 0–80 mm; precision, 0.2 mm). The percentage of body fat was determined using the Slaughter equation. All anthropometrics measurements were performed by the same research team, and each member was specialized in one test, with the aim to reduce possible measurement errors.

### Lower limb strength

The standing long jump test was applied and highest score in centimeters was taken of the two attempts allowed. To perform the test, a tape measure and a non-slippery surface were necessary. This test has been shown to be valid (*Castro-Pinero et al., 2009*) and reliable (*Artero et al., 2011*) with children and adolescents.

### Cardiorrespiratoty fitness

The 20-m shuttle-run test was used to evaluate cardiorrespiratory fitness (*Ruiz et al., 2011*). This Course-Navette is a field-based test, which consisted in running from one line to another line located 20 m away, changing the rhythm in function of a sound signal that increases progressively. The initial speed of the signal is 8.5 km/h, and it increases by 0.5 km/h (1 min equals 1 stage). The test ends when the participant cannot reach the line at the same time as the audio signal for the second consecutive time or when the participant stops due to fatigue. Test performance requires a loudspeaker used to reproduce the sound signal and a 20-m non-slippery surface. The test has been shown to be valid and reliable in children and adolescents (*Artero et al., 2011*; *Castro-Pinero et al., 2009*). We used the Leger formula to calculate the maximal oxygen consumption (cardiorrespiratory fitness, ml/kg/min) (*Léger et al., 1988*).

### Active commuting to and from school

The number of trips and the ACS mode used by students to travel to and from school were calculated with two self-reported questions. The first question asked how they usually traveled to school (i.e., from Monday to Friday) and the second asked how they habitually returned from school (i.e., from Monday to Friday). The validity of these questions has been previously shown to be an adequate method to determine ACS at school in children and adolescents (*Herrador-Colmenero et al., 2014*; *Mora-Gonzalez et al., 2017*; *Muntaner-Mas et al., 2018*). Moreover, participant's information about their home address was asked in the questionnaire. This information was introduced into the Google Earth (*Villa-González et al., 2016b*), as well as the address of the school, with the aim to obtain the distance, time and average slope in the ACS in all participants who reported commuting on foot or bike. It has to be noted that no student reported commuting by bike in this sudy. Finally, the participants were classified in active commuter (number of trips was ≥5, and the time of the trip was ≥15 min), mixed commuter mixed commuter (number of weekly trips was <5, and the time spent on the trip was <15 min) and passive commuter (those who reported traveling regularly by car, motorcycle, or bus) (*Herrador-Colmenero et al., 2014*; *Mora-Gonzalez et al., 2017*; *Muntaner-Mas et al., 2018*).

## Statistical analysis

The characteristics of the study sample are presented as means and standard deviations (SD) or percentages. Furthermore, an analysis of variance for each dependent variable was performed in order to know the differences between boys and girls, and also between active commuter, mixed commuter and passive commuter. Later, also was estimated post-hoc analysis based on Bonferroni Test with the aim to analyze comparisons by pairs.

**Table 1 Descriptive analysis of the study variables and sex differences.**

| | Total | | Boys | | Girls | | $p$ | $t$ |
|---|---|---|---|---|---|---|---|---|
| | M | SD | M | SD | M | SD | | |
| $n_{total}$ | 256 | | 137 | | 119 | | | |
| Age (years) | 13.20 | 0.05 | 13.25 | 1.03 | 13.15 | 0.88 | 0.090 | 0.80 |
| Weight (kg) | 54.11 | 0.74 | 55.61 | 12.10 | 52.31 | 9.79 | 0.079 | 2.37 |
| Tricipital fold (mm) | 17.56 | 0.48 | 16.39 | 8.83 | 19.00 | 6.24 | 0.001 | −2.68 |
| Subescapular fold (mm) | 13.40 | 0.49 | 12.76 | 7.54 | 14.20 | 6.10 | 0.121 | −1.66 |
| Strength of the lower limbs (cm) | 156.28 | 1.87 | 168.05 | 31.26 | 142.12 | 23.53 | 0.052 | 6.93 |
| Number of trips/week | 8.09 | 2.43 | 8.03 | 2.45 | 8.17 | 2.42 | 0.424 | −0.45 |
| Distance (m) | 1,703.50 | 770.62 | 1,732.12 | 860.74 | 1,670.83 | 654.83 | 0.853 | 0.63 |
| Time (min) | 20.44 | 9.24 | 20.78 | 10.32 | 20.05 | 7.85 | 0.853 | 0.63 |
| Average slope (%) | 2.93 | 2.12 | 3.12 | 2.69 | 2.71 | 1.14 | 0.104 | 1.52 |
| Body fat (%) | 26.47 | 0.64 | 25.86 | 11.61 | 27.26 | 6.90 | 0.000 | −1.17 |
| Cardiorrespiratory fitness (mL·kg$^{-1}$·min$^{-1}$) | 41.81 | 0.36 | 43.93 | 5.95 | 39.29 | 4.14 | 0.000 | 6.66 |

Path analysis was used to test the hypothesized model. Robust maximum likelihood (MLR) estimator was used which allows to non-normality of observations and can handle data that are missing at random (*Yuan & Bentler, 2000*). As chi-square ($\chi^2$) values can be inflated and suggest poor model fit with larger samples sizes, the following common goodness-of-fit were considered to assess model fit: comparative fit index (CFI) (*Bentler, 1990*), Tucker-Lewis Index (TLI) (*Bentler & Bonett, 1980*), Root Mean Square Error Of Approximation (RMSEA) (*Steiger, 1990*) and Standardized Root Mean Square Residual (SRMR). Values greater than 0.90 and 0.95 for the CFI and TLI were considered to reflect "reasonable" and "excellent" model fit, respectively, and values smaller than 0.08 or 0.06 for the RMSEA and SRMR were considered to reflect "reasonable" and "excellent" model fit, respectively, based on rules of thumb conventional cut-off criteria (*Browne & Cudeck, 1993*; *Hu & Bentler, 1999*). Finally, in order to test the indirect effects, the path model was re-estimated using bootstrapping resampling procedures ($N = 10,000$) to compute 95% Confidence Intervals (*Preacher & Hayes, 2008*).

## RESULTS

Table 1 shows descriptives of the studied variables and differences by gender. Boys showed lower levels of tricipital fold and body fat, as well as higher scores in cardiorrespiratory fitness when comparing with girls (all, $p < 0.001$).

The hypothesized path model showed an excellent fit to the data: MLR $\chi^2 = 88.316$; df = 6; $p < 001$; CFI = 1.000; TLI = 1.000; RMSEA = 0.000 (90% CI [0.000–0.000]); SRMR = 0.000. Figure 2 presents the direct associations between study variables. Average slope was negatively associated with body fat ($\beta = -0.146$; $p < 0.001$). Body fat was negatively and directly related with cardiorrespiratory fitness ($\beta = -0.281$; $p < 0.001$) and strenght lower limbs ($\beta = -0.371$; $p < 0.001$). However, average slope was not significantly associated with either strenght lower limbs or cardiorrespiratory fitness. As displayed in

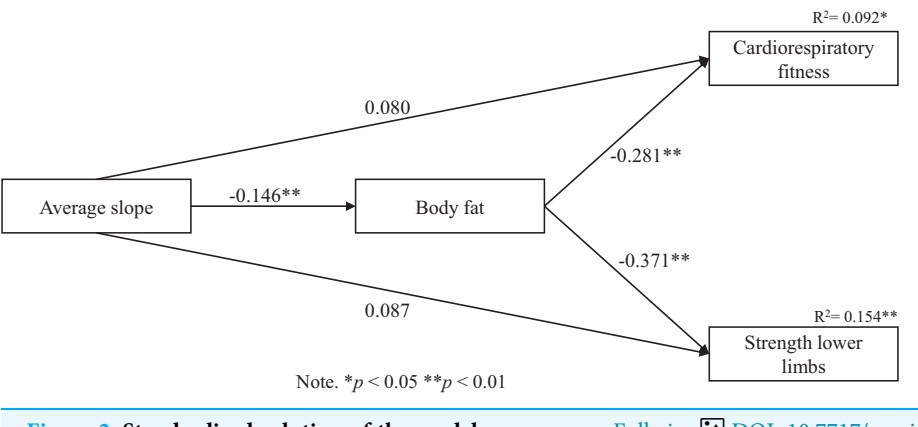

Figure 2 **Standardized solution of the model.**     

Table 2 **Indirect effects.**

|  | β | SE | Bootstrap 95% CI | p |
|---|---|---|---|---|
| Average slope → Strength lower limbs |  |  |  |  |
|    Total effect | 0.141 | 0.075 | [0.018–0.265] | 0.060 |
|    Direct effect | 0.087 | 0.087 | [−0.056 to 0.231] | 0.317 |
|    Indirect effect (via body fat) | 0.054 | 0.036 | [−0.006 to 0.114] | 0.138 |
| Average slope → Cardiorrespiratory fitness |  |  |  |  |
|    Total effect | 0.121 | 0.114 | [−0.067 to 0.309] | 0.290 |
|    Direct effect | 0.080 | 0.135 | [−0.142 to 0.302] | 0.552 |
|    Indirect effect (via body fat) | 0.041 | 0.032 | [−0.011 to 0.093] | 0.195 |

**Note:**
  β, standard coefficient; SE, estimation deviation.

Fig. 2, the model explained 9% of variance from cardiorrespiratory fitness, whereas 15% of of variance from strength of the lower limbs was explained throught the model. Table 2 presents the indirect associations between study variables. Body fat was not found a significant mediator in the association between average slope and physical fitness ($p > 0.05$).

## DISCUSSION

The current study aimed to examine the association between the average slope in the ACS and physical fitness, as well as evaluate the mediator role of fatness in this relathionship. The main finding of this study was the negative association between average slope in the ACS and body fat, which in turn was negatively related with the strength lower limbs and cardiorrespiratory fitness. However, the results did not show the mediating effect of body fat between average slope and cardiorrespiratory fitness and strength lower limbs in the active commuter groups.

In line with our results, many researches have shown a positive relationship between ACS and physical fitness in children and adolescents (*Larouche et al., 2014*; *Martin et al., 2016*). However, few researches have studied the importance of the average slope

during the ACS. In relation with the first hypothesis, results of current research demonstrated a positive association between average slope and strength lower limbs and cardiorrespiratory fitness in active commuter groups. Previous research has confirmed that improving adolescents' cardiorrespiratory fitness and muscle strength affects their cardiovascular health (*Andersen et al., 2011*; *Carson et al., 2016*; *Lubans et al., 2011*). Therefore, previous researchers (*Chillón et al., 2017*) have found the ACS as a way to increase the level of weekly physical activity which in turn is associated with a physical fitness in adolescents.

On the contrary, the studies conducted by *Muntaner-Mas et al. (2018)* and *Villa-González et al. (2015)* found ACS (mainly walking) was not a sufficient stimulation to modify strength lower limbs and cardiorrespiratory fitness in adolescents, with the exception of the strength lower limbs in girls, although these differences were minimal (*Villa-González et al., 2015*). In addition, it is important to note that active commuting to school has been considered an important predictor of physical activity energy expenditure (*Slingerland, Borghouts & Hesselink, 2012*). *Chillón et al. (2017)* found that walking to school implies about more than 7 min of physical activity moderate and vigorous per day/journey in Spanish adolescents, respectively. For this reason, interventions or health policies might consider increasing the intensity of ACS. For instance, promoting commuting by routes with higher average slope as a tool to increase physical activity intensity in order to enhance adolescents' health-related fitness. Researchers should take into account these findings during the ACS because it was showed a positive relationship between average slope to lead the changes in their strength lower limbs and cardiorrespiratory fitness.

In this line, numerous studies have examined specifically how the physical environment affects ACS. The systematic review conducted by *Chillón et al. (2011)* tested that the cross-sectional studies have consistently shown that distance is the strongest predictor of active transportation to school among children, with longer distances associated with lower rates of active commuting. However, few intervention studies have taken into account the average slope. Average slope could be considered as part of the inclusion criteria for intervention studies to target students living within a walkable distance to school. Therefore, according to results, the average slope should be as an inclusion criterion in the ACS.

Regarding the second hypothesis, the results did not confirm that body fat mediates the relationship between the average slope and the strength lower limbs and cardiorrespiratory fitness. Many studies have shown benefits of the ACS, related with reducing body fat, as a way to prevent overweight and obesity of active commuters (*Andersen et al., 2011*; *Muntaner-Mas et al., 2018*; *Noonan et al., 2017*). Despite these results, a negative association was found between body fat and strength lower limbs and cardiorrespiratory fitness in active commuters. In line with these results, the studies conducted by *Mytton et al. (2018)*, *Mytton, Panter & Ogilvie (2016)* and *Poitras et al. (2016)* demonstrated that active travel was associated with reduced visceral adipose tissue, preventing of cardio-metabolic disease. These authors revealed further evidence that

promoting active travel may contribute to improving cardio-metabolic health. In this sense, it is emphasized that adding or including the average slope in active commuter into long-distance commutes is associated with reduced adiposity. Enabling long-distance commuters to do this may require facilities that enable walking or cycling in combination with car-use and public transport.

Finally, in relationship with the results and taking into account the previous literature (*Noonan et al., 2017*; *Villa-González et al., 2015*), this research does not aim to study the differences between active commuter groups, mixed commuter groups or passive commuter. Nevertheless, including the environment elements (i.e., average slope) related with physical fitness in the active commuters as an approach to enhance the knowledge about ACS in adolescents is a purpose to achieve. In this line, numerous studies have examined how the physical environment affects active commuters. The systematic review conducted by *Chillón et al. (2011)* tested that the cross-sectional studies have consistently shown that distance is the important predictor of active transportation to school among children, with longer distances associated with lower rates of active commuting. However, few of the intervention studies included average slope in their study design or analyses, whereas average slope might be considered as part of the inclusion criteria for intervention studies to target students living within a walkable distance to school. Therefore, the average slope was should included as an inclusion criterion, in the same way the distance and time in the ACS to achieve a closer and greater understanding of ACS in previous studies. The results of the present investigation reinforce this idea. Therefore, the main finding of this study is that the average slope to and from school is positively associated with the levels of strength of the lower limbs and cardiorrespiratory fitness of students classified as active commuters. In addition, the mediating effect of body fat in the associations of the average slope to and from school with strength lower limbs and the average slope to and from school with cardiorrespiratory fitness is another important finding.

This study presents some limitations. First, the use of a cross-sectional design does not allow us to establish a causal relationship between the variables. Second, the temporal stability of ACS was not assessed, as it is subject to climatological variables that are very difficult to control, such as a prolonged period of rain, winter cold (*Herrador-Colmenero et al., 2018*). Finally, as was previously indicated (*Garcia et al., 2018*), the biological maturation have a relevant association with physical activity. Despite this issue, the authors recognize this limitation, so individuals' biological maturation (prepubescent or post pubescent) was not assessed, and this can make influence on the development of the physical fitness. The main strengths of this research is the use Google Earth to measure objectively the distance, time and average slope in ACS, showing better precision and accuracy. In addiction, the 20 m shuttle run test was used to measure cardiorrespiratory fitness as a health indicator in adolescents. In the future, it will be interesting to take into account the average slope during ACS, because it has been shown to be associated with the increase of the physical fitness. The implications of present study, highlight the importance of average slope in ACS relationship between physical fitness in adolescents.

Therefore, the average slope should be considered a covariable in the future researches based on ACS.

## CONCLUSIONS

The present research has found positive associations between average slope and the body fat in the ACS. Despite not being able to demonstrate the indirect effect of body fat between average slope and strenght lower limbs, the results are relevant, since it has been possible to confirm a negative association between the body fat and physical condition in active commuter groups.

## ACKNOWLEDGEMENTS

The authors wish to thank the schools, children and their parents who generously volunteered to participate in the study. We also acknowledge all the staff members involved in the fieldwork for their efforts and great enthusiasm.

### Funding

This work was supported by the European Social Fund and Government of Extremadura (Spain) under Grants: TA18027 for D.S.O. and PO17012 for J.J.P. The funders had no role in study design, data collection and analysis, decision to publish, or preparation of the manuscript.

### Grant Disclosures

The following grant information was disclosed by the authors:
European Social Fund and Government of Extremadura (Spain): TA18027 and PO17012.

### Competing Interests

The authors declare that they have no competing interests.

### Author Contributions

- Pedro Antonio Sánchez Miguel conceived and designed the experiments, performed the experiments, analyzed the data, prepared figures and/or tables, authored or reviewed drafts of the paper, and approved the final draft.
- David Sánchez Oliva conceived and designed the experiments, performed the experiments, analyzed the data, authored or reviewed drafts of the paper, and approved the final draft.
- Mikel Vaquero Solís performed the experiments, authored or reviewed drafts of the paper, and approved the final draft.
- J. J. Pulido performed the experiments, prepared figures and/or tables, and approved the final draft.
- Miguel Angel Tapia Serrano conceived and designed the experiments, prepared figures and/or tables, and approved the final draft.

## Human Ethics

The following information was supplied relating to ethical approvals (i.e., approving body and any reference numbers):

The University of Extremadura granted Ethical approval to carry out the study within its facilities (Bioethical Code: 892016).

## Data Availability

Raw data is available in the Supplemental Files.

## Supplemental Information

Supplemental information for this article can be found online at http://dx.doi.org/10.7717/peerj.8824#supplemental-information.

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
