# Peer review of "Relationship between the average slope in the active commuting to and from school and fitness in adolescents: the mediator role of fatness"

_PeerJ, doi:10.7717/peerj.8824_

## Round 0.1 · original submission · Major Revisions

Based on the results of an Appeal by the authors on the initial decision on this manuscript, we are happy to provide a new Major Revision rather than Reject decision. This allows the authors to revise and resubmit but it does not mean that the paper will be necessarily accepted for publication, unless the authors can respond adequately to all three initial reviewers of this manuscript.

· Appeal

Appeal


· · Academic Editor

Reject

We appreciate the hard work that the authors put into designing and writing up the results of the study, but we feel there are some major methodological and presentation issues that do not warrant further consideration in PeerJ.

I would suggest the authors look critically at the comments provided by the reviewers, especially in regards to the lack of citing relevant key studies within this literature and how such literature would then link to the rationale of the study, the potential lack of novelty of this research question and the overall level of written English. Further areas of improvement could include, being more explicit with respect to the measurement error of the primary and predictor variables and how this may influence the confidence in your results, as well as better controlling for aspects of the analysis such as the potential covariance of body fatness and VO2max.

Reviewer 1 ·

Basic reporting

The paper tests an interesting subject in the kinesiology field and many insights can be drawn from this article.

Experimental design

no comment

Validity of the findings

no comment

Additional comments

My concerns about the study are:
1) The article needs to go through a review of the English language, as the text is often difficult to interpret.
2) In studies that concern physical fitness indicators in children and adolescents, biological maturation is a variable that should be considered. The process of growth and development at this stage of life interferes with health-related behaviors and interferes with indicators of physical fitness.
3) What are the sample size calculation parameters? What is the statistical power of the sample employed in this research? Such information should be described.
4) The description of the method is repetitive at many times - example: information on the use of the equation of Slaughter et al appears twice throughout the method, which is not very objective.
5) It is important to describe the technical measurement error of skinfold evaluators.
6) The authors proposed a mediation analysis for each physical fitness (strength and VO2max) variable separately (i.e., Figure 2). This analysis proposal does not contemplate reality, because the indicators of physical fitness manifest simultaneously. Although separate model analysis is important for understanding initial assumptions about a particular topic, it has no practical applications. Thus, it is best to make a model for mediation testing that takes into account both the strength variable and the VO2max variable. In this sense, my suggestion is that the authors test a single mediation model that presents the physical fitness indicators simultaneously. To do this, authors must make use of path analysis (Path Analysis is a special case of structural equation modeling) and test this mediation.

Reviewer 2 ·

Basic reporting

The authors use a clear language of English and a correct article structure.

Major comments:
1)TITLE
The title does not express the real topic of the manuscript. I suggest adding the Google Earth software in the title.
2)INTRODUCTION
It is very large. In general, the introduction have an extension of 2 pages maximum.
Also, I suggest reorganizing the introduction. Start by defining the general problem, and then finish with the specific issue, which is active commuting to school. It has not the opposite way.
The ideas are mixed and unstructured, and the speech is incoherent.
The authors use a lot of references but some of them are not updates.

Minor comments:
1)INTRODUCTION
Line 43: authors use “active school transportation” and in other parts "active transportation to school". I would suggest homogenizing using always “active commuting to school”.
Line 58,75: I suggest updating references.
Line 63: It is not clear if the minutes are per day or per week.

Experimental design

Major comments:
1)METHOD
The title contains "High School" but this word does not appear in the text.
The method is unstructured. I recommend including several information about: numbers of schools in the sample description and recruitment process (contact process, convenience or random schools, all by telephone or how...).
I don't see clear the program that authors applied. The article does not shows information about duration, process or what is it.
The inclusion criteria are very important and in this article does not exist. I recommend including several information about inclusion criteria. Do exist a minimum of informed consent per school?
The author uses somes references to categorize active commuting to school but there are several errors. Firts, I suggest adding all active modes and not only on foot. Then, I recommend searching references because they exist to find another way to categorize active commuting to school.
The author classified "mixed modes" such as: the number of weekly trips was ≤ 4, and the time spent on the trip was < 15 minutes. With this type of classification.. What Do happen with children that commuting to school every day but spent less than 15 minutes because they live near to school? Are not active commuters?
I recommend categorizing active or passive modes only by number of trips and I also suggest categorizing travel time between home and school by reviewing the current bibliography.

Minor comments:
1)METHOD
Line 119: See active commuting p.8 and not page 7.
Line 137: I suggested adding a reference.
Figure 1: I suggest changing the name of the figure. For example: Figure 1. Flow chart of participants according to the mode of travel. Also, I recommend making it more visual. The numbers of trips and time spent is not totally clear.

Validity of the findings

Major comments:
1)CONCLUSIONS
I would suggest including a new paragraph including the future directions of both researching and public health policies.

Minor comments:
Line 315: I suggest adding the means of "AC" during all the text.

Additional comments

Dear authors,

The manuscript does not provide anything new to the literature, since there is a similar work (that it is referenced) and the new inputs (Google Earth software) are weak, not well analysed and organized. Furthermore, there are some major mistakes in the method that are relevant. My opinion is to reject the manuscript. Under my opinion, it does not contribute to the existing literature except an objective measurement of active commuting. However, I provide major and minor comments.

Yours sincerely,

·

Basic reporting

No comment

Experimental design

No comment

Validity of the findings

No comment

Additional comments

Tittle
I suggest some minor changes in the title in order to understand that the “active commuting to and from school” is examined. I would put “Relationship between the average slope in the active commuting to and from school and fitness in adolescents: the mediator role of fatness”
Abstract
Given the aim of the study was to examine the relationships between the average slope in the active commuting to and from school and the strength of the lower limbs and maximal oxygen uptake (VO2max), it would be advisable to change the previous sentence of the aim. For example, “Active commuting to and from school has been recognized as a potential way to improve physical fitness”.
I would suggest changing “the aims of the study tested the” for “This study examined the relationships between…”
I would suggest changing “active commuting to school” for “active commuting to and from school”
I would suggest changing “showed a positive association between the average slope” for “showed a positive association between the average slope in the active commuting to and from school”

Set a point before moreover (i.e., Moreover the results shown a significant indirect effect in the two models).

I would suggest changing “These results are very relevant, as, till now, no studies were found using objective measurements of active commuting like Google Earth Pro software” for “Given to our knowledge, no studies to date have used objective measurements to assess active commuting to and from school, these findings are promising”

Please, rewrite the conclusions. From my point of view, the conclusion are “The average slope to and from school were positive associated…” instead of “Adolescents who actively commuted to school were positive associated with…”

I suggest adding the word “body mass index” or “body fat”, “walking” o “cycling” in the keywords.

Introduction

I would suggest changing “active commuting to school” for “active commuting to and from school” throughout the manuscript.

Given the aim of the study was to examine the relationships between the average slope in the active commuting to and from school and the strength of the lower limbs and maximal oxygen uptake (VO2max), it would be advisable to add in this sentence physical fitness “Active forms of transportation have been recognized as potential ways to increase daily physical activity, providing an alternative to other physical activity domains such as sport and exercise (Heath et al., 2012; Sahlqvist, Song, & Ogilvie, 2012). In addition, I would put “active commuting to and from school” instead of “active forms of transportation” given the context of the study.
For example “Active commuting to and from school have been recognized as potential way to increase daily physical activity or improve physical fitness among adolescents”
I would suggest readind and adding these references in this sentence and throughout the manuscript.

Larouche, R., Saunders, T. J., Faulkner, G. E. J., Colley, R., & Tremblay, M. (2014). Associations between active school transport and physical activity, body composition, and cardiovascular fitness: a systematic review of 68 studies. Journal of Physical Activity and Health, 11(1), 206-227.
Martin, A., Kelly, P., Boyle, J., Corlett, F., & Reilly, J. J. (2016). Contribution of walking to school to individual and population moderate-vigorous intensity physical activity: systematic review and meta-analysis. Pediatric exercise science, 28(3), 353-363.
I would suggest changing the order of some of the paragraphs in the introduction. I would put the following paragraph behind “Active commuting to and from school have been recognized as potential way to increase daily physical activity or improve physical fitness among adolescents”

Previous research has shown that children and adolescents who actively commute to school increase their levels of physical activity between 5 and 37 minutes (Mammen
et al., 2014). Andersen et al. (Lars Bo Andersen et al., 2011) tested the benefits of this increase in children, reporting lower levels of body fat and a lower probability of heart disease. In this regard, Ramírez-Vélez et al. (Ramírez-Vélez et al., 2017) showed that regular active commuting to school may be associated with greater fitness and a lower incidence of metabolic sindrome than using passive transportation, especially in girls. Along with the health risks commonly associated with obesity and overweight (Buckley, Lowry, Brown, & Barton, 2013), there is considerable scientific literature highlighting negative biomechanical changes such as developing unfavourable gait patterns and skeletal misalignments of the lower limbs (Buliung, Faulkner, Beesley, & Kennedy, 2011; Buliung, Mitra, & Faulkner, 2009; Chillón et al., 2011).

I would suggest removing these two sentences “Moreover, walking may be perceived as an easier, safer, and cheaper option, especially for those who are less active. Walking is a more familiar activity, it does not require special equipment and is less likely to involve direct competition for road space with motorised traffic (Morris & Hardman, 1997).”

In the sentence “Different studies have indicated a decrease in the use of active modes of transportation to school during adolescence in many countries (USA, United Kingdom, Australia and Spain) (Chillón et al., 2013; Department for Transport, 2008)” I would suggest adding “Despite the well-known health benefits of active commuting to and from school, different studies have indicated a decrease in the use of active modes of transportation to school during adolescence in many countries (USA, United Kingdom, Australia and Spain) (Chillón et al., 2013; Department for Transport, 2008).

In the sentence “Interest in the influence of environmental factors on adolescents has increased in recent times (Adams et al., 2011; Sallis et al., 2016), valuing different aspects in order to categorize transport as active or inactive (Rodríguez-Rodríguez et al., 2019).” I would put “Due to this decrease, interest in the potential correlates of active commuting to and from school has increased in the last years”

I would suggest readind and adding these references:

Ikeda, E., Stewart, T., Garrett, N., Egli, V., Mandic, S., Hosking, J., ... & Moore, A. (2018). Built environment associates of active school travel in New Zealand children and youth: A systematic meta-analysis using individual participant data. Journal of Transport & Health, 9, 117-131.

Ikeda, E., Hinckson, E., Witten, K., & Smith, M. (2018). Associations of children's active school travel with perceptions of the physical environment and characteristics of the social environment: a systematic review. Health & place, 54, 118-131.
What sections do the authors refer to in this sentence? Which means a “higher positive slope”? How many meters are we talking about? Is there any study about the relationship of the average slope to active commuting to and from school and positive outcomes in youth?

Recently, it has been shown that sections with a higher positive slope are characterized by producing internal mechanical work, involving greater muscular activity and increasing the linear energy demand (Ehrström et al., 2017; Vernillo et al., 2016).

In the sentence “On the other hand, physical activity levels have been widely highlighted as a positive predictor of fitness and a negative predictor of fatness (Janssen & LeBlanc, 2010).” I would suggest adding this reference:

Poitras, V. J., Gray, C. E., Borghese, M. M., Carson, V., Chaput, J. P., Janssen, I., ... & Sampson, M. (2016). Systematic review of the relationships between objectively measured physical activity and health indicators in school-aged children and youth. Applied Physiology, Nutrition, and Metabolism, 41(6), S197-S239.

There is a typographical error in this sentence. For example, Serrano-Sánchez et al. (Serrano-Sánchez et al., 2010) found that adiposity mediated the associations between physical activity and fitness in active adult men.

There is a another typographical error in this sentence. “Additionally, Arngrímsson
and Ólafsdóttir (Arngrímsson & Ólafsdóttir, 2016) demonstrated that % body fat mediated over a third of the relationship between physical activity and cardiorespiratory fitness in late adolescents”.

I would suggest being more cautious in this sentence (see red colour). However, to our knowledge there are not study that evaluated if adiposity could affect in the associations between active commuting and fitness components.

In the aims I would put “average slope to and from school”

In the hypotheses, I would put “average slope to and from school” instead of “the average slope during active commuting to the school”

In the hypotheses, I would put “…to achieve higher levels of cardiorespiratory
fitness.”

Method

Design and Participants

Authors point out “The participants selected were those who enrolled in the Program to Promote Physical Activity and Healthy Habits in Adolescents”.

- Please put all words in lower case as it is not the name of a program.
- Was it a school-based intervention or a extracurricular intervention? Please, indicate in the text

Please, rephrase this sentence. “The present investigation was a cross-sectional by conglomerate”.

How many schools did the study sample come from?
Please put “active commuters” instead of “active”.

Please put “final sample” instead of “reference sample”.

In the “design and participants” section, authors said “For the analysis, only those students classified as Active (see Active Commuting p. 7) were taken as a reference
sample”. However, in the “instruments” section authors said “The students’ classification according to the type and number of trips and distance to the school was: Active, Mixed, or Inactive (Mora-Gonzalez et al., 2017)” Therefore, did only active commueters participate in this study? Please, clarify this issue.

Procedure

Please put these four sentences together “All gave their informed consent for inclusion before they participated in the study. … Participation was voluntary and confidential. Written informed consent was obtained from parents and participants. Students who did not present the informed consent were excluded from the study”

Instruments

Authors point out “The number of trips and the transportation mode used by students
travel to and from school were calculated with two self-reported questions (Herrador-Colmenero, Pérez-García, Ruiz, & Chillón, 2014; Mora-Gonzalez et al., 2017; Muntaner-Mas et al., 2018; Saelens & Handy, 2008)” What were those two questionnaires? There are four references.

Authors point out “The first question asked how they habitually traveled to school and the second asked how they habitually returned from school.” I would put (See red colour) “The first question asked how they habitually traveled to school each day of the week (i.e., from Monday to Friday) and the second asked how they habitually returned from school each day of the week (i.e., from Monday to Friday)”

Please, rephrase this sentence “A total of ten weekly trips were established if they reported walking to and from the institute and zero trips if they reported traveling by car, bus, or motorbike”

Please, put “Aerobic capacity” instead of “Aerobic Capacity”
Please, put “Active commuting” instead of “Active Commuting”.

Please, change the name of this study variable “Classification as Active, Mixed, or Inactive.”. For example, active commuter, mixed commuter, and passive commuter.
Please change the name of this study variable throughout the article to avoid misunderstandings.

Covariates

Which is CRF and CA? Please, clarify it. In many parts of the text the authors use “active commuting” and in anothers sections “AC”. Please, unify this word.

Results

Please, put “active commuters” instead of “active students” in this sentence “Table 1 shows the characteristics of body composition, fitness, and the active
transportation to school of the active students”

Please, change “active subjects” for “active commuters”

In addition, authors said “The students’ classification according to the type and number of trips and distance to the school was: Active, Mixed, or Inactive (Mora-Gonzalez et al., 2017)” and “These classification criteria were based on the findings of (Martínez-Gómez et al., 2011; Ruiz-Ariza, de la Torre-Cruz, Redecillas-Peiró, & Martínez-López, 2015; Villa-González, Ruiz, et al., 2016) (Figure 1)” What is the reference for choosing these criteria from all of these studies? Please, clarify it.

Given that the authors excluded students who reported bicycling as an active means of transport, I wonder if it wouldn't be more precise put “walking to and from school” instead of “active commuting to and from school” throughout the manuscript.

Please rephrase this sentence. “The results of the active subjects did not reveale significant differences between to sex for age, weight, subscapular fold, strength of the lower limb, number of trips/week, distance, time and average slope (all p > 0.05). Among which groups were there no differences in the study variables?

Please add “significantly” in this sentence “However, the girls had significantly higher values of the tricipital folds, % body fat and VO2máx than the boys (all p < 0.01).

Put “between” instead of “netween”

Put “active, mixed and inactive commuters” instead of “active, mixed and inactive
groups”.

Please, put “participants” or “students” or “adolescents” instead of “subjects”

Discussion

It is important to try to justify some of the results found in the discussion.

Please remove “during actively commuting to school” in the first hypothesis.
Please remove “when actively commuting” in the second hypothesis.

I would put “the average slope to and from school” instead of “the average slope during active commuting to the school”

In the figure 1, I would put “active commuter, mixed commuter, and passive commuter” instead of “active participants, etc.”

I would put “the average slope to and from school” instead of “the average slope covered”.

Please remove “when actively commuting to school” in this sentence “To date, the present study is the first research that associates the average slope of the route when actively commuting to school with the physical and anthropometric characteristics of active adolescents.”

There is a typographical error in this sentence “Chillón et al. (Chillón et al., 2011)”

Authors said “The contributions of the present investigation contrast with most of the prior research collected in the review by Chillón et al. (Chillón et al., 2011; Muntaner-Mas et al., 2018), who confirmed that most investigations carried out to date have focused on using active commuting to school as a means of promoting physical activity, without evaluating the physical or anthropometric benefits that this could generate for the students” Please, review this systematic review and reformulate this sentence.

Larouche, R., Saunders, T. J., Faulkner, G. E. J., Colley, R., & Tremblay, M. (2014). Associations between active school transport and physical activity, body composition, and cardiovascular fitness: a systematic review of 68 studies. Journal of Physical Activity and Health, 11(1), 206-227.
Please consider putting this paragraph in another section. For example, to highlight strengths of this study. “Several studies have pointed out the lack of objectivity and quantification of the instruments used in most research assessing active commuting (Batista et al., 2018; Chillón et al., 2011; Muntaner-Mas et al., 2018). Taking into account these limitations, we decided to use Google Earth Pro software, an instrument whose validity and reliability has been previously confirmed (Villa-González, Rodríguez-López, et al., 2016).”

Authors said “As far as we know, there is no obvious association between AC and cardiorespiratory fitness among children and adolescents (Villa-González et al., 2015).”
Please, review this reference to reformulate this paragraph.

Larouche, R., Saunders, T. J., Faulkner, G. E. J., Colley, R., & Tremblay, M. (2014). Associations between active school transport and physical activity, body composition, and cardiovascular fitness: a systematic review of 68 studies. Journal of Physical Activity and Health, 11(1), 206-227.
There is a typographical error in this reference (L. B. Andersen, Lawlor, Cooper, Froberg, & Anderssen, 2009; Østergaard, Kolle, Steene-Johannessen, Anderssen, & Andersen, 2013).

There is a typographical error in this reference Villa-González et al. (Villa-González et al., 2015)

In this sentence “In this regard, Villa-González et al. (Villa-González et al., 2015) were not able to find significant associations between AC and physical fitness variables such as cardiorespiratory fitness and lower muscular fitness for boys” I would put “…were not found significant…”

Please, change “as a polluting variable” for “as a contaminating variable”

There are two sentences in the discussion section that are very similar “In relation to these results, our study does not attempt to find differences between active, mixed and inactive groups, since it is more than demonstrated (Villa-González et al., 2015)” and
“Taking into account the previous literature, this research does not intend to study the differences between active, mixed or inactive students”. Please, reconsider remove one of them.

Please, reformulate this sentence “Taking into account the mediating effect of body fat on the association of the average slope with the strength lower limbs and average slope withVO2max”

In this sentence “Therefore, our study concludes that the average slope presents a positive association between the levels of strength of the lower limbs and VO2máx of students classified as active. In addition, the mediating effect of body fat in the associations of the average slope with streght lower limbs and the average slope with VO2máx.” I would put “Therefore, the main finding of this study is that the average slope to and from school is positively associated with the levels of strength of the lower limbs and VO2máx of students classified as active commuters. In addition, the mediating effect of body fat in the associations of the average slope to and from school with strenght lower limbs and the average slope to and from school with VO2máx is another important finding”

In this sentence “The temporal stability of the habit of actively commuting was not assessed,” I would put “The temporal stability of active commuting to and from school…”

Please put a reference in this limitation “The temporal stability of the habit of actively commuting was not assessed, as it is subject to climatological variables that are very difficult to control, such as a prolonged period of rain, winter cold...”

For example, Herrador‐Colmenero, M., Harrison, F., Villa‐González, E., Rodríguez‐López, C., Ortega, F. B., Ruiz, J. R., ... & Chillón, P. (2018). Longitudinal associations between weather, season, and mode of commuting to school among Spanish youths. Scandinavian journal of medicine & science in sports, 28(12), 2677-2685.

Conclusions

Please, rewrite the conclusions. From my point of view, the conclusion are “The average slope to and from school were positive associated…” instead of “Adolescents who actively commuted to school were positive associated with…”

I would put “These results are promising because there are no studie to date that have examined the relationships of the study variables using and objective measurement of active commuting such as the Google Earth software” instead of “These results are very relevant because, to date, no studies were found that used an objective measurement of active commuting such as the Google Earth software.”

---

## Round 0.2 · Minor Revisions

I thank the authors for their major efforts in attending to the comments of the reviewers on the initial version of the manuscript. Please attend to the final remaining comments so to maximise the chance of your paper being accepted for publication in PeerJ.

·

Basic reporting

I want to thank you again for the opportunity to review the Manuscript ID: Relationship between the average slope in the active commuting to and from school and fitness in adolescents: the mediator role of fatness (#41013). This manuscript is an excellent contribution to the literature on the relationship between the average slope in the active
commuting to and from school and fitness in adolescents.

I was surprised by the quality of the authors’ revision effort. Frankly, I thought the reviewers were asking too much of the authors to handle/revise, but the author team carried out an excellent revision. Specifically, the authors have strengthened the introduction and discussion section. My personal recommendation is to accept the manuscript with minor changes.

Experimental design

Nothing to add.

Validity of the findings

Nothing to add.

Additional comments

I want to thank you again for the opportunity to review the Manuscript ID: Relationship between the average slope in the active commuting to and from school and fitness in adolescents: the mediator role of fatness (#41013). This manuscript is an excellent contribution to the literature on the relationship between the average slope in the active
commuting to and from school and fitness in adolescents.

I was surprised by the quality of the authors’ revision effort. Frankly, I thought the reviewers were asking too much of the authors to handle/revise, but the author team carried out an excellent revision. Specifically, the authors have strengthened the introduction and discussion section. My personal recommendation is to accept the manuscript with minor changes.
Abstract
1.- I would suggest changing “test the relationships between the average slope and physical fitness,…” for ““test the relationships between the average slope in the active commuting to and from school and physical fitness,…”.

2.- I would suggest changing “were assessed” for “participated in this study”.
3) I would suggest changing “Specifically, in the active commuter,” for “Specifically, in the active commuter group”.
4) I would suggest changing “which in turn was related” for “which in turn was positively related”.
Introduction
1.- I would suggest changing “Active Commuting (AC) can be defined….” for “Active commuting to and from school (ACS) can be defined…”. I would also change “AC” for “ACS” throughout the article.
2.- Please remove this sentence because the idea is repeated in the above sentence “Previous research has shown that adolescents who actively commute to school can increase each day their levels of physical activity between 5 and 37 minutes (Mammen et al., 2014)”.
3.- Please change “AC to and from school” for “ACS”

4.- Please, change the order of these two references (Mytton, Panter & Ogilvie, 2016; Carson et al., 2016).

5.- How many hypotheses are in the study? Two or four? It is not clear.

Derived from these aims, 1) the main hypotheses were: the average slope during AC to and from school contributes to develop physical fitness, and 2) to achieve better levels of cardiorespiratory fitness. In accordance with the aims indicated, several hypotheses are suggested. The first hypothesis suggested that the active commuters would have a better physical fitness. The second hypothesis proposed that the active commuters would have a lesser fatness and as a consequence would have a better physical fitness.

Method

Design and Participants

1.- Some parts in the “Sample section” are written in future tense (e.g., The invitation to participate in the study will be made by draw). I would use past tense (e.g., The invitation to participate in the study was made…)
2.- Add the name of your university because it not a blind review.
Instruments
1.- I would change the name of the variable “active commuting” for “active commuting to and from school”.
2.- Please, change “on foot and bike” For “on foot or bike”
3.- Please change this sentence “However, the analyzes do not change because no participant reported go cycling commuter to school” for “It has to be noted that no student reported commuting by bike in this study”.
4.- Line 140. Please, change the order of these two references: Castro-Piñero et al., 2009; Artero et al., 2011

5.- Line 164. Add “groups” in this sentence “between active commuter, mixed commuter, and passive commuter groups”.


Results

1.- Line 194. Please, add “p” in italics.

Discussion
1.- Line 203-204. Add “groups”. “However, the results did not show the mediating effect of body fat between average slope and cardiorespiratory fitness and strength lower limbs in the active commuter group”
2.- Line 205-206. Authors said, “In line with our results, many researches have shown a positive relationship between AC to and from school in children and adolescents (Larouche et al., 2014; Martin et al., 2016), …”.
a) The name of a variable is missing from this sentence. For example, “…a positive relationship between ACS and …”? physical fitness?
3.- Line 203-204. Add “groups”. “association between average slope and strength lower limbs and cardiorespiratory fitness in active commuter group”.
4.- Please, change the order of these three references (Lubans et al., 2011; Andersen et al., 2011; Carson et al., 2016).
5.- Please, add a reference to support this claim “Therefore, previous researchers have found the AC to and from school as a way to increase the level of weekly physical activity which in turn is associated with physical fitness in adolescents”
6.- Line 218. What does mean “although these differences were minimal”? Was there or was there not a significant difference?
7.- Authors said “Researchers should take into account these findings during the AC to and from school because it was showed a positive relationship between average slope to lead to lead the changes in their strength lower”
a) Please, remove “to lead” because it appears twice.
8.- 257-265. Remove these sentences. They appear twice in the discussion.
9.- Please, change “The main strengths of the research are…” for “The main strength of this research is…”
10.- Line 290. Add “groups” in this sentence “active commuter group”.
11. Please put "cardiorespiratory" instead of "cardiorrespiratory" throughout the manuscript.

Conclusions

1.- Please remove this sentence “These results are promising because there are not studies to date that have examined the relationships of the study variables using and objective measurement of active commuting such as the Google Earth software” and add the main implications of these findings.

---

## Round 0.3 · Minor Revisions

All the reviewers and I are satisfied with the work you've put into improved this manuscript. Please address the very minor typographical errors identified by the reviewer so that we can then accept this paper for publication in PeerJ.

·

Basic reporting

I want to thank you again for the opportunity to review the Manuscript ID: Relationship between the average slope in the active commuting to and from school and fitness in adolescents: the mediator role of fatness (#41013). This manuscript is an excellent contribution to the literature on the relationship between the average slope in the active
commuting to and from school and fitness in adolescents.

My personal recommendation is to accept the manuscript.

I'm just pointing out some typos.

1.- Please, put in the abstract and method section the same number of trips.
Abstract: mixed commuter (number of weekly trips was ≤ 4, and the time spent on the trip was < 15 minutes)
Method: mixed commuter (number of weekly trips was < 5, and the time spent on the trip was < 15 minutes)
2.- Please, unify AC and ACS throughout the manuscript.

3.- Please, put these references in the right order.

(Noonan et al., 2017; Muntaner-Mas et al., 2018)

(Villa-González et al., 2015; Muntaner-Mas et al., 2018)

4. Please check the final references. There are some journals where the full name does not appear.

Experimental design

No comments in this section.

Validity of the findings

This manuscript is an excellent contribution to the literature on the relationship between the average slope in the active commuting to and from school and fitness in adolescents.

Additional comments

I want to thank you again for the opportunity to review the Manuscript ID: Relationship between the average slope in the active commuting to and from school and fitness in adolescents: the mediator role of fatness (#41013). This manuscript is an excellent contribution to the literature on the relationship between the average slope in the active commuting to and from school and fitness in adolescents.

My personal recommendation is to accept the manuscript.

I'm just pointing out some typos.

1.- Please, put in the abstract and method section the same number of trips.
Abstract: mixed commuter (number of weekly trips was ≤ 4, and the time spent on the trip was < 15 minutes)
Method: mixed commuter (number of weekly trips was < 5, and the time spent on the trip was < 15 minutes)
2.- Please, unify AC and ACS throughout the manuscript.

3.- Please, put these references in the right order.

(Noonan et al., 2017; Muntaner-Mas et al., 2018)

(Villa-González et al., 2015; Muntaner-Mas et al., 2018)

4. Please check the final references. There are some journals where the full name does not appear.

---

## Round 0.4 · accepted · Accept

Thanks for addressing the final small required edits.